# Untargeted Metabolomics Reveals a Complex Impact on Different Metabolic Pathways in Scallop *Mimachlamys varia* (Linnaeus, 1758) after Short-Term Exposure to Copper at Environmental Dose

**DOI:** 10.3390/metabo11120862

**Published:** 2021-12-11

**Authors:** Vincent Hamani, Pascaline Ory, Pierre-Edouard Bodet, Laurence Murillo, Marianne Graber

**Affiliations:** UMR 7266 LIENSs, CNRS-La Rochelle Université, 2 Rue Olympe de Gouges, 17000 La Rochelle, France; pascaline.ory01@univ-lr.fr (P.O.); pierreedouard.bodet@univ-lr.fr (P.-E.B.); laurence.murillo@univ-lr.fr (L.M.); marianne.graber@univ-lr.fr (M.G.)

**Keywords:** copper, metabolomics, UHPLC/QToF mass spectrometry, scallop

## Abstract

Ports are a good example of how coastal environments, gathering a set of diverse ecosystems, are subjected to pollution factors coming from human activities both on land and at sea. Among them, trace element as copper represents a major factor. Abundant in port ecosystem, copper is transported by runoff water and results from diverse port features (corrosion of structures, fuel, anti-fouling products, etc.). The variegated scallop *Mimachlamys varia* is common in the Atlantic port areas and is likely to be directly influenced by copper pollution, due to its sessile and filtering lifestyle. Thus, the aim of the present study is to investigate the disruption of the variegated scallop metabolism, under a short exposure (48 h) to a copper concentration frequently encountered in the waters of the largest marina in Europe (82 μg/L). For this, we chose a non-targeted metabolomic approach using ultra-high performance liquid chromatography coupled to high resolution mass spectrometry (UHPLC-HRMS), offering a high level of sensitivity and allowing the study without *a priori* of the entire metabolome. We described 28 metabolites clearly modulated by copper. They reflected the action of copper on several biological functions such as osmoregulation, oxidative stress, reproduction and energy metabolism.

## 1. Introduction

Trace elements contamination of soil and water is a recurrent environmental problem that requires increased monitoring and is the subject of much worldwide research. Among these heavy metals, copper (Cu) seems particularly interesting to study because it is present in most compartments (soil, air and water) [1,2,3], and impacts many organisms (animals, plants and bacteria) [4,5,6].

Copper, as many other trace elements, has paradoxical roles: it is necessary for many metabolic activities [7,8,9] and becomes toxic when present at too high levels [8,10,11]. For example, the copper is an essential cofactor for many enzymes [7,12] and it is necessary for oxygen transport in the haemolymph of many crustaceans and bivalves [13,14]. However, copper at higher concentrations can affect enzyme activity, antioxidant defence [15] and haemocyte survival [10]. Nguyen et al. show that mortality of mussel (*Perna canaliculus*) haemocytes significantly increases as soon as the amount of copper reaches 62.5 μM (i.e., 1 g/L), whereas it was almost absent at 25.0 μM (i.e., 0.4 g/L) [10]. Similarly, Viera et al. showed that the enzymatic and antioxidant activities in the fish *Pomatoschistus microps* are affected by copper at concentrations between 25 to 200 μg/L [15].

Copper is a common trace element in port areas. Indeed, due to human activities and their location along the coasts, ports are exposed to metal contamination coming from corrosion of structures, carburant, antifouling paints, and run-off water [16,17]. Especially, their shape tends to amplify the concentrations of these elements trapped in this semi-closed environment, as the structure of the ports is made in a way to limit the effect of waves and currents [18,19]. In an analysis conducted in 2018 in the Europe’s largest marina (Les Minimes, France), among the 8 trace elements tested, copper was the most common element after zinc reaching a concentration of up to 82 μg/L in the water column of the Marillac basin, one of the four port basins [12].

Ports, although often ignored as such, are ecosystems in their own right [20,21,22]. As well as being the site of significant human activities, they are biodiversity hotspots and provide refuge, food and nursery for many organisms like most coastal environments [21,22]. For public health and environmental reasons, it seems legitimate to ask what effect copper has on the organisms, at concentration levels found in this environment. As such, *Mimachlamys varia*, commonly known as the variegated scallop, which is a filter-feeder bivalve frequently found in the marine regions of the Atlantic coast, is often used for marine biomonitoring, particularly in the Minimes marina [23,24,25,26]. Because of its capacity to bio-accumulate trace elements such as copper and because of its sensitivity to these elements it appears to be a relevant bio-indicator species [25,26,27,28,29,30]. Bustamante et al. highlighted that *Pectinidae* such as the variegated scallop are of particular interest for biomonitoring programs, due to their high bioaccumulation potential compared to other bivalves [23]. Furthermore, Milinkovitch et al. demonstrated the value of the variegated scallop as an indicator of environmental pollution, by showing that biomarkers of oxidative stress (Malondialdehyde and superoxide dismutases (SOD)) and a marker of the immune system (Laccase-type Phenoloxidase activity) present in this species responded efficiently to the presence of trace elements such as copper [25]. Similarly, Breitwieser et al. showed that the variegated scallop could be used as a relevant bio-indicator by finding a correlation between concentrations of trace elements such as copper and indicators of oxidative stress, immune system impairment, altered mitochondrial respiration or modified enzyme activity like phosphatases [27]. Breitweiser et al. highlighted that scallops were more sensitive bioindicators than mussels and oysters, as they were the only ones having a marked enzymatic response (SOD, glutathion S-transférase and laccase) to trace elements in the same environment [30]. Two other studies also showed that the genetic diversity of variegated scallops was significantly reduced when trace elements were present in the environment, which further reinforces the relevance of this model for biomonitoring purposes [26,27].

However, all the studies presented above are experiments conducted in situ and the relationship between the organism and the pollution is carried out for all the trace elements present on the site. These studies did not discriminate the effect of one trace element in relation to another and, to our knowledge, no study to date has shown the effect of copper on the biomonitoring model, the variegated scallop. In addition, complete molecular mechanism of copper-induced toxicity in bivalves is far from being well known, although previous studies have started to elucidate it through metabolomics approach. These studies have shown that copper induces responses in energy metabolism, glycerophospholipid metabolism, oxidative stress, osmoregulation and apoptosis mechanisms in bivalves such as mussels (*Perna canaliculus*), Manila clams (*Ruditapes philippinarum*) and estuarine oysters (*Crassostrea hongkongensis*) for concentrations of 1 g/L (3 h copper-exposure on haemocytes), 40 μg/L (96 h copper-exposure) and 50 μg/L (2–6 weeks copper exposure) respectively [10,31,32]. These responses are related to the modulation of metabolites involved in energy storage (α, β-glucose, ATP/ADP and glycogen), the transsulfuration pathway (cysteine and methionine), the glutathione metabolic pathway (cysteine, glutamic acid and glutathion) and in the Krebs cycle (succinate, citrate and α-ketoglutarate) [10,31,32]. All these studies highlight the fact that, due to the low fraction of metabolites identified, similar studies seem necessary to further investigate and clarify the effects of copper on the metabolism of bivalves.

The aim of the present study is to have a deeper insight of the effect of copper, at the concentration maximally found in the environment of Europe’s largest marina (82 μg/L), on the metabolic pathways of a model organism that strongly bioaccumulates trace elements: *Mimachlamys varia* (Linnaeus, 1758).

In order to focus only on the effect of copper, a laboratory experiment was conducted on 78 scallops. These scallops were collected from a site considered as a reference site in previous studies, due to the low presence of trace elements [27,33]. Half of the scallops collected were exposed (48 h) to a copper dose of 82 μg/L and the others were used as controls. Efficient extraction method followed by comparative metabolomic profiling of gill samples allowed determining how the metabolic profile of scallop gills is modified, in response to copper exposure. More generally, untargeted metabolomics approach is used to measure the widest range of metabolites present in an extracted sample, without prior knowledge of the metabolome. It can reveal new, non-targeted and early detected dysregulated features. Thus, in the present paper, an optimized triple extraction method followed by an untargeted metabolomics approach using ultra-high performance liquid chromatography coupled to high resolution mass spectrometry (UHPLC-HRMS) was chosen, as it offers a high level of sensitivity to highlight the impacts of short-term exposure to copper. After a complete processing of analytical data and statistical analysis of results using multivariate techniques, we could clearly separate scallops with no metal inputs and those exposed to copper and identify about thirty copper-modulated metabolites.

## 2. Results

### 2.1. LC/MS Data Processing and Analyses

XCMS preprocess from Workflow4Metabolomics platform (W4M) resulted in the detection of 21,483 and 35,880 *m*/*z* features for negative and positive ionization modes, respectively. Data process implying blank removal, loess batch correction of analytical drift and removal of metabolites with Coefficient of Variation (CV) > 0.3 in Quality Control samples (QC = pool of all samples) led to 1925 and 5997 ions in negative and positive ionization modes, respectively. Multivariate analyses were performed on these latter datasets. Principal Component Analyses (PCA) highlighted not only the natural structure of samples but also potential sample outliers (Figure 1). On the score plot, the PCA showed the sample distribution based on the qualitative and quantitative metabolites composition. Variable metabolite intensities induced a clear clustering into two sample groups according to the copper contamination, in negative as well as in positive ionization mode. Indeed, metabolite composition contributed to the well separated structure of samples between control and copper exposed individuals, mainly along the first axis (t1) for positive ionization mode and along the second axis (t2) for negative mode. The first two axes (t1 and t2) accounted for 11–17% and 9–10% of the total variability for positive and negative modes, respectively. Outlier detection relies on the score distance represented by Hotelling T2 threshold and on orthogonal distance. These two parameters showed that three reference samples (two T2 and one orthogonal distance) in negative mode and two reference samples (one T2 and one orthogonal distance) in positive mode were outliers. These samples were then removed before Partial Least Square-Discriminant Analysis (PLS-DA), because of their high level in one or both distances.

PCA analysis allowed detecting a clear natural and distinguished clustering between control and copper-exposed samples, justifying PLS-DA model reliability. This is a supervised method building a model that force the distinction between two firstly defined groups. PLS-DA also provides results of variable responsible for forced clustering. The relevance and performance of the supervised built model was proven with PLS-DA parameters of data consistency (R^2^) and prediction performance (Q2). R^2^ (cumulative) reached 0.99 for both ionization modes and Q2 (cumulative) reached 0.84 and 0.89 for positive and negative modes, respectively. Permutation test (n = 100) and cross-validation test provided *p*-values < 0.01 confirming the consistency of the data and the reliability of the predicted models. Thus, a separation between the two groups was significantly demonstrated.

Among variables structuring the sample distribution of PLS-DA model, Variable Importance in Projection (VIP) > 1 defined the ones with the significant contribution to the model. We kept metabolites with VIP > 1 to rebuild new successive PLS-DA models, until reaching the best predictive model and select the most important metabolites explaining the two-group clustering. In positive mode, two supplementary models were built, keeping 1587 and 487 metabolites with VIP > 1 and Q2 reaching 0.97. In negative mode, only one new model was needed to obtain the best predictive model (492 metabolites with VIP > 1 and Q2 = 0.95).

### 2.2. Metabolite Modulation

Among ions significantly modified under copper exposure, a total of 28 metabolites were annotated with a score of less than or equal to 3 in Shymanski scale (Table 1). Two of them have a confirmed structure (score 1), 15 a probable structure (12 score 2a and 3 score 2b) and 11 are potential candidates (score 3) (Table 1). They were all significantly different between copper-exposed and control samples, mainly because they were selected by their variable importance in projection (VIP) after successive PLS-DA modeling. Among them, 14 ions were annotated in negative ionization mode, 12 in positive ionization mode and two were annotated in both ionization modes (Table 1).

They belong to different biochemical classes: 3 are carbohydrates, 10 are peptides or amino acids (AAC), 13 are lipids and one is a nucleotide (Figure 2). Interestingly, most of them were up-regulated in copper-exposed samples compared to the control samples, and only 4 out of 28 were down-regulated (three amino acids and one lipid) (Figure 2). The highest magnitude of modulation after copper-exposure was 31-fold relative spectral intensity for N5-Acetyl-N2-gamma-L-glutamyl-L-ornithine (up-regulated) and 13-fold relative spectral intensity for oxidized gluthatione (down-regulated).

## 3. Discussion

One of the most significant challenges, when performing environmental untargeted metabolomics study, is identifying which metabolic pathways are altered, in relation with the metabolite variations observed in stressor-exposed organisms. Many metabolites are involved in several pathways and are the product or substrate of many different enzymes or processes, in particular for central metabolic pathways, where any one metabolite play a role in a myriad of pathways. The translation from changes in metabolites relative abundance to physiological interpretation is therefore a key issue. Nevertheless, in the following discussion, we propose different hypotheses on the physiological effects connected with the putatively identified metabolite that present a significant modulation after copper exposure. At this stage, it is not possible to establish connections between the potentially involved metabolic pathways, nor to prioritize the effects. It is clear that these hypotheses, based on previous works, need further experiments to be supported, but they show the complexity of copper toxicity mechanisms and provide a solid starting point for further biological interpretation.

Scallops underwent alterations of their metabolism after 48 h exposure to a nominal copper concentration of 82 μg/L in laboratory conditions. These alterations concerned about 30 compounds belonging to all families of biochemical compounds. These variations reflect potential effects on several of the biological processes discussed below, such as energy metabolism, oxygen transport, inflammation, defence against oxidative stress, lipid metabolism, osmoregulation, reproduction, peptide and nucleotide metabolism.

### 3.1. Energy Metabolism and Oxygen Transport

It is commonly accepted that exposure to pollutants leads to an additional energy cost for marine organisms, which must expend energy to implement cellular protective and detoxification mechanisms [34]. Moreover, toxic metals such as cadmium, copper, zinc and mercury have been shown to reduce both mitochondrial efficiency and coupling and increase proton leakage in marine organisms [34].

In response to this additional need for ATP, these organisms are able to switch between different metabolic processes involved in energy acquisition and conversion, in particular towards anaerobic pathways [34]. So, the first arising question is whether recourse to the latter pathways is observed here. The aspartate-succinate pathway with succinate and alanine as end products is one of the four main anaerobic pathways used by marine bivalves [35]. Therefore, if anaerobic metabolism occurs, a decrease in aspartate is expected. Such a decrease is indeed observed in this case, which would therefore confirm the use of anaerobic metabolism by scallops exposed to Cu. Cao and collaborators also observed a decrease in several amino acids including aspartate, when studying copper-induced metabolic variation in oysters [36]. They attributed this phenomenon to a mobilization of amino acids to produce energy for maintaining function of pathways in these Cu-exposed organisms. On the contrary, Zhang and collaborators found an increase of aspartate level in gill tissues of Cu-exposed clams after exposures for 24 h at 10 or 40 μg/L Cu^2+^ [31]. This was interpreted as a mean for marine molluscs of balancing their intracellular osmolarity with the environment, with high intracellular concentrations of free amino acids.

In the present study, the increase of four oligosaccharides (tri-, tetra-, penta- and hexa-saccharides) in Cu-exposed scallops was found and may be related to a disturbance in the catabolism of glycogen. In marine bivalves, energy reserves are mainly constituted by glycogen and to a lesser extent by lipids. Glycogen is the storage form of glucose and plays a central role in the energy supply for both maintenance metabolism and gametogenesis [37,38].

Glucose supply from glycogen requires synchronous activities of two different enzymes: glycogen phosphorylase and the bifunctional glycogen debranching enzyme. Glycogen phosphorylase cleaves the α-1,4 linkage from the non-reducing end of the glycogen to remove glucose 1-phosphate. Further, when four glucose residues remain before the branching point, the bifunctional debranching enzyme goes into action. This enzyme exhibits both glycosyltransferase and amylo-1,6-glucosidase activities on a single polypeptide chain [39]. Using the transferase activity, the debranching enzyme transfers the three glucose residues from the four-residue glycogen branch to a nearby branch. This leaves one glucose unit joined to the glucose chain branch point through an α-1,6-glycosidic linkage.

After that, through the glucosidase activity, the debranching enzyme cleaves the α-1,6 linkage to release a free glucose from the branch point. Glycogen phosphorylase is then able to continue the cleavage of glucose residues from this linearized glycogen chain, producing glucose 1-phosphate. This last sugar is then converted to glucose-6-phosphate and enters glycolysis and tricarboxylic acid cycle [40]. These two enzymes responsible for glycogen breakdown have been isolated in clams [41,42]. One hypothesis that can be put forward to explain the accumulation of tri-, tetra-, penta- and hexa-saccharides upon copper exposure is that the glycogen debranching enzyme may undergo partial inactivation by copper. Such an inactivating effect of copper has been described in mussels for the hexokinase enzyme [43]. The results of in vivo experiments, after 3 days exposure to Cu^2+^ at 40 μg/L, indicated that the inhibition of hexokinase would be caused, on the one hand, by a direct binding of the metal to the sulfhydryl groups of the protein, and on the other hand, more indirectly, by a decrease in the level of reduced glutathione (GSH) and consequent imbalance between pro-oxidant and antioxidant cellular processes. Indeed, GSH is used by metal exposed organisms to protect themselves against their cytotoxicity [44].

To our knowledge the susceptibility of the bivalve glycogen debranching enzyme to copper has not been studied but should be examined to confirm the hypothesis expressed above. An early study shows that the rabbit glycogen debranching enzyme could be inactivated by oxidation of its sulfhydryl groups promoted by the oxidized form of GSH [45].

A variation of two fatty acid derivatives was found here: a down-regulation of adipoyl-CoA and an up-regulation of 3-hydroxyoctadecenoylcarnitine. We already observed variations of carnitine combined to fatty acids levels in previous studies about the effect of emersion periods and Zn exposure on the variegated scallop [11,35]. The fatty acid concerned here (3-hydroxyoctadecenoic) is a long one, which has to be combined with carnitine to enter the mitochondria, where it undergoes beta-oxidation for energy production. So, its increase reflects a disturbance in the oxidation of the corresponding long-chain fatty acids upon Cu exposure. However, as mentioned by the Connor and collaborators, metabolism of acylcarnitines is not only related to the transport of fatty acids, but also plays a key role, among others, in maintaining the homeostasis of the mitochondrial acyl-CoA/CoA ratio [46,47]. Moreover, here, an additional effect related to fatty acid metabolism is observed with the decrease in the level of adipoyl-CoA. So, it seems that a more general disturbance of fatty acids metabolism occurs in presence of Cu, with potential mitochondrial dysfunction.

In mollusks, oxygen transport is provided by haemocyanin, a multimeric glycoprotein containing a type-3 copper [48]. This respiratory protein was reported to decrease in crustaceans in response to exposition to polluted environment including copper. The hypothesis was that detoxification of contaminants may interfere with normal copper metabolism in the blue crab and may lead to a decrease in the synthesis of haemocyanin [49]. As branched chain amino acids like leucine are involved in haemocyanin synthesis and transportation [50], the increased level of leucine found in the present study may be linked to a decrease of the respiratory protein upon copper exposure. This trend was observed by Cao and collaborators when studying the combined effect of salinity and copper exposure in oysters [36]: An increased level of leucine was observed in 15 g/kg seawater plus Cu compared with 10 g/kg plus Cu waters.

### 3.2. Inflammation, Oxidative Stress Defence and Lipid Metabolism

Lipids have a variety of biological roles as fuel molecules or energy store [51]. They are also essential constituents of biological membranes that can regulate their functions [52,53,54]. Previous studies showed that copper exposure induced disturbances in the metabolism of lipids in bivalves [55,56].

The diglyceride (9M5/9D3/0:0) or (11M3/9D3/0:0) or (9D3/11M3/0:0) or (9D3/9M5/0:0) up-regulated 5 times under exposure to 82 μg/L of copper for 48 h consists in two fatty acid chains covalently bond to a glycerol molecule through ester linkages. This molecule is an intermediate in the biosynthesis of phosphatidylcholine or phosphatidylethanolamine [57,58]. These phospholipids are among the most abundant in the soft tissues of bivalves, in contrast to phosphatidylserine, which is a quantitatively minor membrane phospholipid [58]. In our study, the compound putatively identified as phosphatidylserine was up-regulated by a factor 3 in copper-exposed samples. The increase of this compound in the presence of copper was also reported in bivalve mollusk *Anadara broughtonii* [58]. During the early stages of apoptosis, the externalization of phosphatidylserine to the outside of cells is considered as a recognition signal, by which apoptotic cells are eliminated by phagocytes [59]. The externalization of phosphatidylserine is, in this respect, used as an indicator of apoptosis in a study on the mussel *Mytilus galloprovincialis* [59]. Thus, this increase in phosphatidylserine and diglycerides reflects a membrane modification that could result from an apoptotic process allowing the organism to eliminate cells impacted by copper or containing copper.

The compound putatively identified as leukotriene, up-regulated in the present study by a factor 2 in presence of Cu, could be leukotriene C4, a cysteinyl leukotriene or 11-trans-leukotriene C4, a leukotriene derivative. Leukotrienes are a family of potent inflammatory mediators and natural lipids that are oxygenated metabolites of arachidonic acid [60]. Biosynthesis of the leukotrienes involves the action of a lipoxygenase on arachidonate to yield an intermediate, which is then dehydrated to the leukotriene A4. Leukotriene A4 can be hydrolyzed to the leukotriene B4 or it can be conjugated with glutathione (GSH) to produce the leukotriene C4 [60]. In case of tissue damage, release of leukotriene C4 pro-inflammatory mediator might be involved in apoptosis mechanisms via oxidative stress [61]. This supports the previously stated hypothesis that the organism appears to eliminate cells impacted by copper or containing copper cells by an apoptotic process.

In the present study, an increase of phosphatidic acid level in gills of copper-exposed bivalves was observed. Phosphatidic acid is an important component in phospholipid synthesis. More precisely, it is a precursor for both cytidine diphosphate diacylglycerol and diacylglycerol. Cytidine diphosphate diacylglycerol is used to make phosphatidylinositol, phosphatidylglycerol, and cardiolipin [54]. Thus, phosphatidic acid is indirectly an essential component of membranes from a structural and functional point of view [58]. Phosphatidic acid plays a key role in the regulation of intracellular membrane transport through poorly understood mechanisms. Both phosphatidic acid and lysophosphatidic acid may influence biophysical properties of the membrane [62]. The regulation of phosphatidic acid could be correlated with the membranous modification generated by the mechanisms of apoptosis or endocytosis stated above. This would support this hypothesis.

As mentioned above, phosphatidic acid is a substrate for the enzyme cytidine diphosphate diacylglycerol-synthase to produce an endoplasmic reticulum pool of cytidine diphosphate diacylglycerol [54], another metabolite up-regulated in the present study after copper exposure. Cytidine diphosphate diacylglycerol is a critical intermediate in the synthesis of both lipids, phosphatidylinositol and phosphatidylglycerol/cardiolipin [63,64]. Both molecules, found in bivalves [58], have specialized roles in cells and are synthetized in different organelles.

N-palmitoyltaurine was found to be up-regulated by a factor 3 in Cu-exposed scallops. This compound is part of N-acyl amides, a group of endogenous lipids, characterized by a fatty acyl group linked to a primary amine metabolite by an amide bond. This type of compounds has been found in all organisms from bacteria to mammals for decades. N-acyltaurine were identified as cell-signaling molecules, but many other roles remain to be elucidated [65].

The lysophospholipids compounds putatively identified as lysophosphatidylcholine and lysophosphatidylethanolamine were up-regulated by a factor 3 in gills of scallops upon exposure to 82 μg/L of copper for 48 h. Lysophospholipids are usually the result of phospholipase A-type enzymatic activity on regular phospholipids such as phosphatidylcholine or phosphatidic acid, although they can also be generated by the acylation of glycerophospholipids or the phosphorylation of monoacylglycerols. Phospholipases A are present in bivalves. For example, in Sajiki and Taguchi’s study, the major metabolites from phosphatidylcholine in oyster and scallop were lysophosphatidylcholine and free fatty acid [66]. Lysophosphatitylcholine is a precursor of lysophosphatidic acid, another lysophospholipid that has important signaling functions, especially in inflammation [67]. Lipidomic analysis by Chan and Wang conducted in digestive gland of oyster *Crassostrea hongkongensis* exposed to Cu showed different responses depending on the exposure dose (10 and 50 μg/L for 1, 2 and 4 weeks) [32]. This study suggest that exposure induced subcellular compartmentalization, modulation of inflammatory responses and phospholipid remodeling with rapid elevation in glycerophospholipids like phosphatidylcholine and phosphatidylethanolamine to protect the vital metabolic function of the organ.

Ceramide phosphoethanolamine (d15:2(4E,6E)/20:0(2OH)) putatively identified in our study was found to be up-regulated by a factor 4 under copper exposure. This compound, a sphingomyelin analog, is a major sphingolipid in invertebrates [68]. Ceramide phosphoethanolamine biosynthetic mechanisms differ from sphingomyelin ones, due to the invertebrate specific ceramide phosphoethanolamine synthase. For example, ceramide phosphoethanolamine appears to have a role in early development of *Drosophila* and in axonal ensheathment by disrupting the tight packing of lipid membranes [69].

Metals, and in particular copper, are able to induce oxidative stress in bivalves. Cu^2+^ generates free radicals through the Fenton and Haber-Weis reaction [44]. Cu^2+^-exposed haemocytes showed a remarkable increase in reactive oxygen species production, which induced oxidative stress in mussel *Perna canaliculus* [10]. This was accompanied by a decrease in GSH. This compound is an important antioxidant, able to react with oxidants like H_2_O_2_, with conversion of two GSH molecules into its oxidized form (GSSG). Surprinsingly in our case, GSSG was found to be down-regulated in gills of scallops upon Cu exposure. These results are in accordance with a study conducted in 2000 on oysters which indicated that the copper-induced decrease in glutathione content was mainly related to a stimulation of GST activity, which was transient in the gills [70].

S-(Formylmethyl)glutathione is also modulated by copper. This ubiquitous oligopeptide is described in the literature but its role in scallop metabolism remains to be determined [71].

### 3.3. Osmoregulation, Reproduction

Prostaglandins (PGs) are an important group of bioactive lipid compounds, involved in a great variety of physiopathological processes, like for instance inflammation, signaling or reproduction and furthermore in the control of gametogenesis, ion transport, and defence in marine invertebrates [72]. In the present study, we found that three compounds, putatively identified as PGs or PG-glutathione complexes, were modulated in the gills of scallops exposed to 82 μg/L of copper for 48 h. A metabolite annotated as PG D1, E1, F2 or H1 or as 8-isoPG F2 or E1 showed an up-regulation (x3), a compound putatively identified as S-(PGA1)-glutathione or S-(9-deoxy-12-PGD2)-glutathione or S-(11-OH-9-deoxy-9,12-PGD2)-glutathione was down-regulated (x6) and another compound putatively identified as S-(9-deoxy-delta9,12-PGD2)-glutathione or S-(PGJ2)-glutathione or S-(PGA2)-glutathione was up-modulated (x2). These variations potentially correspond to disturbance of biological functions such as reproduction, ion transport and defence caused by copper in scallops. PGE2 was shown to be involved in sodium transport regulation in gills of the freshwater mussel *Ligumia subrostrata* and in clams [73] and it was shown that hyposmotic stress significantly increased PG synthesis in the marine bivalve *Modiolus demissus* [74]. This is the second time that PG-glutathione conjugates are described in clams, showing that these compounds are indeed present in bivalves and that they are affected by metals [11].

### 3.4. Peptide and Nucleotide Metabolism

In the present study, exposure to 82 μg/L of copper for 48 h led to the modulation of four dipeptides (Val-Asp, HydroxyPro-Tyr, Pro-Arg and N5-Acetyl-N2-gamma-L-glutamyl-L-ornithine), one oligopeptide (glycylalanylprolylmethionylphenylalanylvalinamide) and one amino acid (N-Acetyl-L-glutamate 5-semialdehyde). Except for HydroxyPro-Tyr, these peptides were up-regulated. Dumas et al. also showed an increase of peptides, in the presence of pharmaceutical pollutants, but without finding an explanation of why these particular peptides were modulated [75].

Impact of pollution on nucleotide metabolism of marine mollusk or fish has often been mentioned [11,75,76,77] Nucleotides are the building blocks of DNA and RNA and the metabolic relationships between nucleotides and many metabolic pathways are strong, which makes it difficult to interpret their levels of variation. In the present study, an increase of 2-Hydroxyadenine (Isoguanine) level was found and may be, for example, linked with ATP alteration [55,56,57,58,59,60,61,62,63,64,65,66,67,68,69,70,71,72,73,74,75,76,77,78].

## 4. Materials and Methods

### 4.1. Experimental Design and Sample Preparation

The experimental design is based on an experiment conducted by Ory and collaborators [11]. For the present study, 78 scallops (size between 4–6 cm) were collected at La Pointe du Grouin in Loix en Ré along the French Atlantic coast (Ré Island, France). This site is considered as a reference site in other studies due to the low presence of trace elements [27,33]. A genetic approach was applied in a previous study to validate the identity of the species collected at this reference site [26]. This study also confirmed that the scallop *M. varia* along the French Atlantic and English Channel coasts is not genetically subdivided. Consequently, the scallops found at the refence site are comparable to those found in the port of Les Minimes [26]. The scallops were then acclimated during 4 weeks before the experimentation. The physico-chemical parameters of the water were: pH 8.0, temperature 10 °C, salinity 33 g/kg seawater, nitrite, nitrate and ammonia <0.025 mg/L. These parameters were monitored every day and kept constant by a biweekly water renewal. Shellfish Diet 1800^®^ (Reed Mariculture, Campbell, CA, USA) was supplied *ad libitum* three times a week. Scallops were distributed in 6 aquaria of 18 L. Three aquaria randomly selected were contaminated with a solution of CuSO_4_ (nominal copper concentration 82 μg/L). After 48 h exposure, all 39 scallops from each condition were collected. A 48-h exposure time was established in order to observe the effects of copper after the longest possible exposure, but before lethal effects occur. In a pre-test study, high mortality in scallops was observed after 72 h of exposure to copper at the same concentration (82 μg/L, maximal concentration measured in study site), while the controls were still alive (personal observation). Thus, to study the effects of copper (82 μg/L) on the metabolism of scallops one day before lethal effects occur, we chose an exposure time of 48 h. This choice is reinforced by a study showing that copper can have lethal effects in some bivalve species after 48 h of exposure to lower concentrations than in the present study [79].

Each sample was prepared from the gills of 3 scallops, which were dissected, dried and snap-frozen in liquid nitrogen as in the publication by Ory and collaborators [11]. Thus, a total of 13 control and 13 copper-exposed samples were obtained. Gills in bivalves such as scallops were enlarged during evolution and are much more extensive than needed for respiration. They form the main interface between the organism and the surrounding water and have become therefore a key organ for food absorption. Given the high water filtration rates of bivalves, gills constitute also a significant pathway of incorporation of pollutants and among them metals via seawater [80]. Making a logical extension to this, it has been shown that gills would have high defense ability against contaminants, which are characterized by induction of pollutant biotransformation and antioxidant-related enzymes and metallothioneins [81]. Consequently, gills play a central role in scenarios of acute exposure to metals, by integrating both absorption and metabolism.

The samples were then crushed on ice and adjusted to 1 g. To extract a maximum of compound, each sample was subjected to a triple acetone/acetone/methanol extraction as previously described [35,82]. The three solvent supernatants from each sample were then pooled. To remove residual impurities, they were centrifuged at 3000× *g* for 5 min. This total supernatant was recovered and dried under a stream of nitrogen as previously described [35,82]. The dry extract was finally resuspended in 2 mL of 20/80 methanol/water and filtered at 0.2 μm (using low protein binding filter) before MS analyses [35,82]. The methanol and acetone used were of HPLC grade purity (CARLO ERBA Reagents, Val-de-Reuil, France).

### 4.2. UHPLC/QToF MS Analysis of Samples

An ultra-high performance liquid chromatography (“Acquity UPLC H-class”, Waters, Milford, CT, USA) coupled to high resolution mass spectrometry equipped with an electrospray ionization source was used to analyze the samples (“XEVO-G2-S Q-TOF”, Waters, Manchester, UK). 5 μL of the samples were injected in a column “Acquity UPLC HSST3” (Waters) (2.1 × 150 mm, 1.7 μm), and the products were eluted at a flow rate of 200 μL/min using a gradient composed of solvents A (water/formic acid 100/0.001 (*v*:*v*)) and B (acetonitrile/formic acid 100/0.001 (*v*:*v*)), according to the following procedure: 0–3 min, 100% A; 3–8 min 0–50% B; 8–13 min 50% B; 13–20 min 50–95% B; 20–30 min, 95% B, 30–31 min 95–0% B, 31–36 min 100% B. The analyses were performed in positive and negative ionization mode with MS function in a centroid mode. For the two ionization mode the MS parameters was applied in the ESI source were: source temperature 120 °C, desolvation temperature 500 °C, gas flow-rate of the cone 50 L/h, desolvation gas flow-rate 300 L/h, and capillary voltage was 3 kV (+) and 2.5 kV (−) and sampling cone 35 V. The instrument was adjusted for the acquisition on a 50–2100 *m*/*z* interval, with a scan time of 0.5 s. A 20 to 40 V ramp was used as collision energy for the high energy mode of MS^E^ (Waters). This MS^E^ approach consists in MS and MS/MS data acquisitions in a single same run. This is achieved by rapidly alternating between two functions i.e., the first, acquired at low energy provides exact mass precursor ion spectra; the second, at elevated energy provides high energy exact mass of the fragment ions. Fragment ion spectra are assigned to their associated precursor ion peaks so that all the information necessary to identify each compound of interest is collated and available. Software algorithms that profile each chromatographic peak and determine their retention times accomplish this. Precursors and fragment spectra are then aligned according to retention times and linked together. The Leucine Enkephalin (M = 555.62 Da, 1 ng/μL) was used as a lock-mass and the mass spectrometer was calibrated using 0.5 mM sodium formate solution.

The samples were analysed randomly to avoid the effect of possible analytical drift. Analytical repeatability was guaranteed by quality control samples (QC) that were injected every 5 measurements. The QCs were obtained from the pooling of all samples. Blanks prepared with the last extraction solvent were injected at the beginning and end of the sample sequence to subtract components from the extraction solvent.

### 4.3. Statistical Analysis

The data were processed as ion peak intensity using the Workflow4Metabolomics (W4M) platform (http://workflow4metabolomics.org, accessed on 10 December 2021) according to the method described by Ory and collaborators [35]. All the parameters used for the treatment are available at this link: https://workflow4metabolomics.usegalaxy.fr/u/vhamani/h/imported-test-batch-cuivre-ngatif (accessed on 10 December 2021).

Analytical drift was corrected on the pools, using a Loess regression model [83]. Repeatability was assessed through the coefficient of variation (CV) of the QCs. Metabolites with a CV > 0.3 were removed from the analyses.

Principal component analysis (PCA) was used to detect natural clustering between samples and to detect outliers. Outlier samples were removed to strengthen the model [84]. Partial least squares-discriminant analysis (PLS-DA) was performed on the log-transformed and Pareto normalized data. The selection of metabolites was based on the importance of their contribution as a predictor variable in the PLS-DA model. A data with variable in projection (VIP) > 1 can be considered as a metabolite having a significant contribution to the PLS-DA model. The predictive performance and reliability of PLS-DA models was tested with premutation (n = 100) and cross-validation (10-fold) tests.

To statistically prove the intensity difference of each metabolite between samples, t-tests were performed using the software R (Version 3.6.3 “Holding the Windsock”) with a rejection threshold of 5%. The homoscedasticity of the data and the normality of their distribution was checked beforehand by Shapiro and Bartlett tests.

### 4.4. Metabolite Annotation

The metabolites were annotated using the hmdb (http://www.hmdb.ca, accessed on 10 December 2021), metlin (https://metlin.scripps.edu/landing_page.php?pgcontent=mainPage, accessed on 10 December 2021) and lipidmaps (http://www.lipidmaps.org/tools/ms/LMSD_search_mass_options.php, accessed on 10 December 2021) databases. Searches were performed on the mass values of the metabolites, with an allowed error inferior to 5 ppm. Metabolites were expressed in relative spectral intensity calculated for each metabolite by dividing the spectral intensity in each sample by the median spectral intensity in all samples. Analytical standards were used and analyzed according to the same method as explained above (LC/MS) to verify the identification of leucine and aspartic acid (Sigma-Aldrich, Darmstadt, Germany). Comparison of these standards (retention times, *m*/*z* and fragments) with the QC validates the identification of these ions (Figure 3).

Finally, we used the classification of Shymanski et al., to support the identification of the metabolites [85]. This method assigns a score to each metabolite based on the degree of confidence in its identification [85]:-Score 1, identification using a standard (same retention times, *m*/*z* and fragments).-Score 2a, annotation using fragmentation data from databases such as HMDB, LipidMaps, Metlin, with a spectrum-structure match unambiguous.-Score 2b, the fragments obtained match completely with the proposed structure, which excludes other possibilities, but the data are not completely available in the databases.-Score 3, proposed annotation of one or more isomeric molecules without the possibility of distinguishing between them because few or no fragments were obtained, or the fragments were common to the different positional isomers.

Polysaccharides with different structures cannot be distinguished because they have the same formula and identical fragment ions. Thus, polysaccharides have been grouped under generic names corresponding to the number of monosaccharides that compose them.

## 5. Conclusions

This study constitutes a broad survey on the metabolites impacted by copper exposure in scallops, at environmental concentration (82 μg/L). We used an optimized triple extraction method and as analytical technique, ultra-high performance liquid chromatography coupled to high resolution mass spectrometry (UHPLC-HRMS), to detect as many metabolites as possible, without *a priori*. However, given the variety of chemical classes and physical properties that characterize metabolites, it is clear that not all the different chemical classes of metabolites could be covered in a single analytical method. The significant varying metabolites between non-exposed and exposed organisms have been isolated using multivariate statistics and among them, nearly 30 have been identified or putatively identified. The result is that we have a partial view of copper-induced metabolic shifts. Under these conditions, we can only put forward hypotheses on the metabolic pathways affected, which remain to be confirmed. Furthermore, at this stage, it is not possible to establish connections between the different metabolic pathways, nor to prioritize the effects and to elucidate the complete molecular mechanism of copper toxicity, the complexity of which is already highlighted by our results.

Besides these limitations of an untargeted metabolomics approach, this study demonstrates some of the strengths of using this method: (i) the detection of ‘distress signals’ at the molecular level, faster than in organism/population/ecosystem level biomarkers (ii) the identification of metabolic changes that would be missed by other ‘omics’ techniques, as protein and mRNA levels in the cell do not necessarily translate into metabolite changes (iii) the detection of unexpected varying metabolites, showing that short exposure to copper at environmental condition leads to significant changes in the metabolism of scallops, some of which are signs of cellular damage.

Further studies are needed to formally identify the key metabolic pathways involved in copper toxicity. Tools are available which attempt to confirm assertion made for altered pathways through (i) subsequent focus on key pathways of interest with a targeted metabolomics analysis, (ii) an enrichment analysis, (iii) the building of networks to help interpret the data, (iv) isotopic labelling a substrate and following this label through various metabolic pathways.

## Figures and Tables

**Figure 1 metabolites-11-00862-f001:**
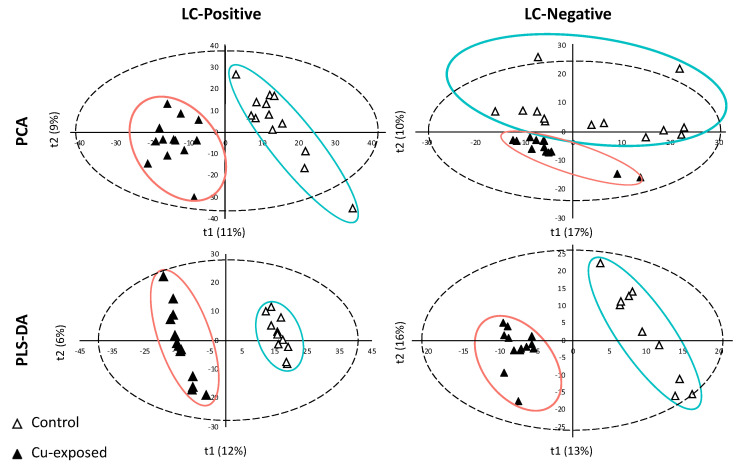
Score plots of PCA and PLS-DA for positive mode (LC-Positive), negative mode (LC-Negative) of liquid chromatography. The dotted ellipse represented the confidence limit (95%) of Hotelling’s T2 statistic. The control samples and the Cu-exposed samples are visually grouped in green and red ellipses, respectively.

**Figure 2 metabolites-11-00862-f002:**
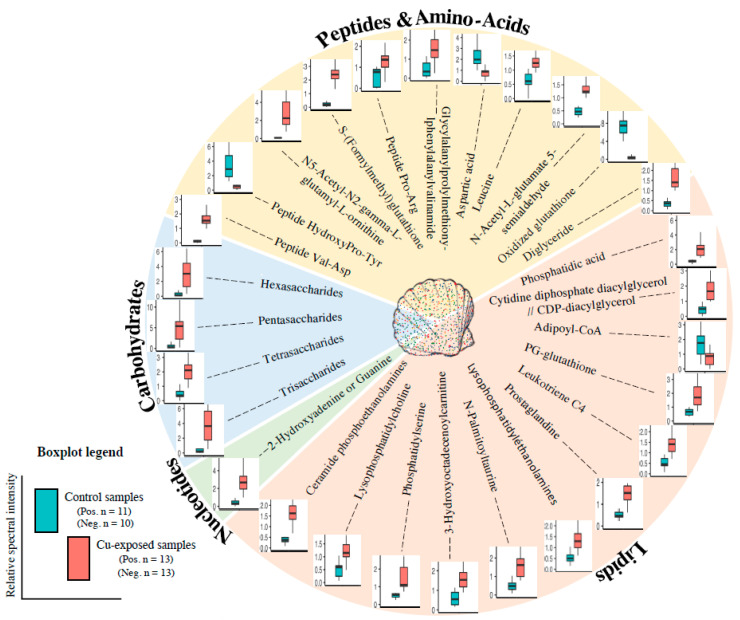
Metabolites found in the gills of scallop showing a significant difference between control and copper-exposed samples (48 h at 82 μg/L) classified by family and box plots of their relative spectral intensity.

**Figure 3 metabolites-11-00862-f003:**
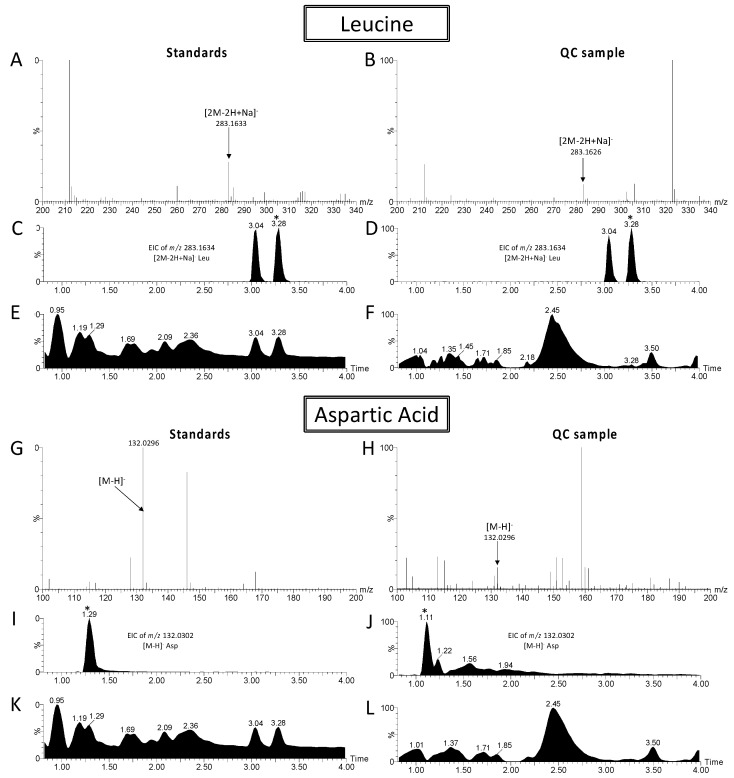
Comparison of chromatograms and mass spectrometry spectra between Standards (left) and QC samples (right) for identification of leucine and aspartic acid. Mass spectrometry spectra for the leucine standard (**A**), for the leucine in QC sample (**B**), for the aspartic acid standard (**G**) and for the aspartic acid in QC sample (**H**). Extracted Ion Chromatograms (EIC) for the leucine standard (**C**), for the leucine in QC sample (**D**), for the aspartic acid standard (**I**) and for the aspartic acid in QC sample (**J**). Total ion current chromatograms for the leucine standard (**E**), for the leucine in QC sample (**F**), for the aspartic acid standard (**K**) and for the aspartic acid in QC sample (**L**). * Peaks of interest.

**Table 1 metabolites-11-00862-t001:** Metabolites varying after 48 h copper exposure (82 μg/L). The score represents the Shymanski classification. The power of the modulation (Copper effect) is measured as the difference between the relative spectral intensity means.

	Metabolite	Mode	Retention (min)	Formula	MonoisotopicMass (Da)	Adduct	Observed Mass (*m/z*)	Theoretical Mass (*m/z*)	Mass Error (ppm)	Score	Copper Effect
Carbohydrate	Trisaccharides	Pos	1.45	C_18_H_32_O_16_	504.1690	[M + H]^+^	505.1777	505.1763	2.8	2a	 × 8
Tetrasaccharides	Pos/Neg	1.46 /2.52	C_24_H_42_O_21_	666.2219	[M + H]^+^	689.2123/665.2133	667.2291	1.5/2	2a	 × 4
Pentasaccharides	Pos	4.64	C_30_H_52_O_26_	828.2747	[M + H + K]^2+^	434.1180	434.1226	10.5	2a	 × 10
Hexasaccharides	Pos	5.72	C_36_H_62_O_31_	990.3275	[M + H + K]^2+^	515.1465	515.1490	4.9	2a	 × 10
Peptide & AAC	Peptide Val-Asp	Pos	3.28	C_9_H_16_N_2_O_5_	232.1059	[M + H]^+^	233.1136	233.1132	1.6	2a	 × 8
Peptide HydroxyPro-Tyr	Pos	4.16	C_14_H_18_N_2_O_5_	294.1216	[M + H]^+^	333.0862	333.0847	4.5	2a	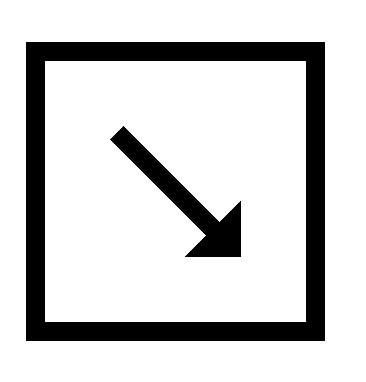 × 6
N5-Acetyl-N2-gamma-l-glutamyl-l-ornithine	Pos	4.64	C_12_H_21_N_3_O_6_	303.1430	[M + H]^+^	304.1507	304.1503	1.3	2a	 × 31
S-(Formylmethyl)glutathione	Pos	6.39	C_12_H_19_N_3_O_7_S	349.0944	[M+Na]^+^	372.0856	372.0836	5.2	2b	 × 10
Peptide Pro-Arg	Pos	8.06	C_11_H_21_N_5_O_3_	271.1644	[M + H]^+^	310.1288	310.1276	4.0	2a	 × 2
Glycylalanylprolylmethionylphenylalanylvalinamide	Pos/Neg	8.27	C_29_H_45_N_7_O_6_S	619.3152	[M+2H]^2+^/ [M − H]^−^	310.6649/618.3034	310.6649/618.3079	0/7.3	2b	 × 3
Aspartic acid	Neg	1.11	C_4_H_7_NO_4_	133.0375	[M − H]^−^	132.0295	132.0302	5.5	1	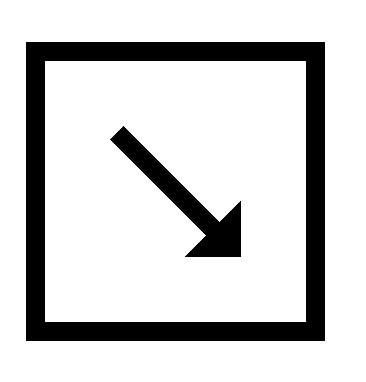 × 3
Leucine	Neg	3.3	C_6_H_13_NO_2_	131.0946	[2M − 2H + Na]^−^	283.1629	283.1634	1.7	1	 × 2
N-Acetyl-l-glutamate 5-semialdehyde	Neg	5.74	C_7_H_11_NO_4_	173.0688	[M − H]^−^	172.0608	172.0615	4.3	2a	 × 2
Oxidized glutathione	Neg	6.12	C_20_H_32_N_6_O_12_S_2_	612.152	[M − H]^−^	611.1436	611.1447	1.8	2a	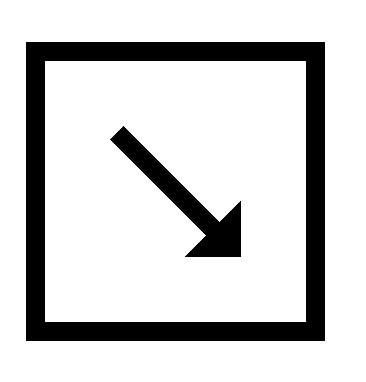 × 13
Lipid	Diglyceride (9M5/9D3/0:0) or (11M3/9D3/0:0) or (9D3/11M3/0:0) or (9D3/9M5/0:0)	Pos	6.49	C_40_H_66_O_7_	658.4809	[M + H + K]^2+^	349.2269	349.2256	3.8	3	 × 5
Phosphatidic acid	Pos	7.11	C_41_H_69_O_8_P	720.4730	[M + H + K]^2+^	380.2225	380.2217	2.1	3	 × 5
Cytidine diphosphate diacylglycerol or CDP-diacylglycerol (CDP-DG)	Pos	7.48	C_45_H_83_N_3_O_15_P_2_	967.5299	[M+H+Na]^2+^	495.7663	495.7632	6.3	3	 × 4
Adipoyl-CoA	Neg	6.81	C_27_H_44_N_7_O_19_P_3_S	895.1626	[M − 2H]^2−^	446.5629	446.5740	24.8	2a	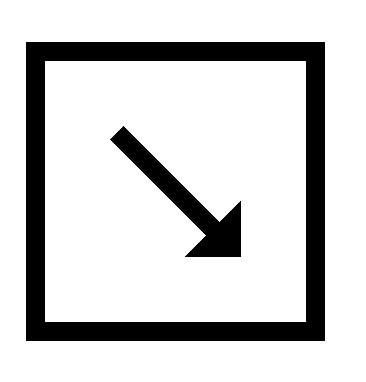 × 2
S-(PGA1) or S-(9-deoxy-D12-PGD2) or S-(11-OH-9-deoxy-D9,12-PGD2) or S-(9-deoxy-delta9,12-PGD2) or S-(PGJ2) or S-(PGA2)-glutathione	Neg	8.58	C_30_H_47_N_3_O_10_S	641.2982	[M − H]^−^	640.2889	640.2909	3.1	3	 × 3
Leukotriene C4 or 11-trans-Leukotriene C4	Neg	10.3	C_30_H_47_N_3_O_9_S	625.3033	[M − H]^−^	624.2950	624.2960	1.6	3	 × 2
Prostaglandin D1, E1, F2 or H1 or as 8-isoprostaglandin F2 or E1	Neg	12.88	C_20_H_34_O_5_	354.2406	[M − H]^−^	353.2325	353.2333	2.3	3	 × 3
Lysophosphatidyléthanolamines (lysoPE) (P-16:0/0:0) and (0:0/18:2(9Z,12Z))	Neg	17.31/ 17.36	C_21_H_44_NO_(6/7)_P	437.2906/477.2855	[M − H]^−^	436.2816/476.2769	436.2833/476.2782	4/2.8	3	 × 2
N-Palmitoyltaurine	Neg	18.43	C_18_H_37_NO_4_S	363.2443	[M − H]^−^	362.2360	362.2371	3.1	2a	 × 3
3-Hydroxyoctadecenoylcarnitine	Neg	18.7	C_25_H_47_NO_5_	441.3454	[M + K − 2H]^−^	478.295	478.294	2.1	3	 × 3
Phosphatidylserine 20:5(5Z,8Z,11Z,14Z,17Z) or 18:3(9Z,12Z,15Z) or 18:4(6Z,9Z,12Z,15Z) or 20:4(8Z,11Z,14Z,17Z)	Neg	19.51	C_44_H_70_NO_10_P	824.4495/803.4737	[M + Na − 2H]^−^/[M − H]^−^	824.4495/802.4676	824.4484/802.4665	1.3/1.4	3	 × 3
Lysophosphatidylcholine (LysoPC) (18:2(9Z,12Z)/0:0) or (0:0/18:2(9Z,12Z))	Neg	20.54	C_26_H_50_NO_7_P	519.3325	[M − H]^−^	518.3237	518.3252	2.9	3	 × 2
Ceramide phosphoethanolamines (PE-Cer) (d15:2(4E,6E)/20:0(2OH))	Neg	20.63	C_37_H_73_N_2_O_7_P	688.5155	[M − H]^−^/[M + K − 2H]^−^	687.5066	687.5083	2.5	3	 × 4
**Nucleotide**	2-Hydroxyadenine or Guanine	Pos	4.52	C_5_H_5_N_5_O	151.0494	[M + H]^+^	152.057	152.0567	2.0	2b/3	 × 7

## Data Availability

Data is contained within the article.

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
