# Peer review of "Untargeted Metabolomics Reveals a Complex Impact on Different Metabolic Pathways in Scallop *Mimachlamys varia* (Linnaeus, 1758) after Short-Term Exposure to Copper at Environmental Dose"

_metabolites, 2021, doi:10.3390/metabo11120862_

Round 1

Reviewer 1 Report

  1. Author explained that the VIP value was derived after removing the outlier. Why did the author remove the outlier, and is this reasonable? In addition, there are still outliers in the figure.
  2. Author noted that several identified metabolites were verified as the standard. However, there is no information related to the targeted analytical method in the manuscript. 

Author Response

We thank you for your comments and for their relevance. We have tried to provide as much clarification as possible in the text to guide readers on the questions you asked.

You will also find a more detailed response to your questions below.

  1. Author explained that the VIP value was derived after removing the outlier. Why did the author remove the outlier, and is this reasonable? In addition, there are still outliers in the figure.

To perform multivariate methods like PLS-DA, data must be previously preprocessed. Among scaling and transformation, the data also need to be cleaned before determining how well the PLS-DA model fits with the data. Cleaning includes the detection of outlier and the removal of the strong ones that would disturb a model. Hotelling’s T2 and orthogonal distance of PCA analysis provides strong outliers. That’s why we chose to remove the outliers detected from PCA results before PLS-DA. We added a specification about this comment in a manuscript (Lines 560-562).

The figure 1 (PCA plots) shows the whole samples (exposed vs. unexposed) and shows that despite the very large number of ions in each sample, there is a significant difference between the exposed and unexposed samples. However, due to the large number of ions in each sample, it is unlikely that they are all the same even for the same treatment. It is therefore normal to observe some outliers. This figure allows to see how different the samples can be and to act if necessary (lines 127-128: "Principal Component Analyses (PCA) highlighted not only the natural structure of samples but also potential sample outliers"). Thus, Figure 1 shows the PCA (top) with outliers and the PLS-DA (bottom) after correction.

  1. Author noted that several identified metabolites were verified as the standard. However, there is no information related to the targeted analytical method in the manuscript. 

The standards were analyzed in the same way and under the same conditions as the samples (LC/MS). Then, the ions were compared to the results of the analysis of these standards (same m/z, retention time and fragments). The spectra corresponding to the analyzed standards were added in the material and method, as well as a sentence explaining the way we analyzed the standards (line 578-582 & Figure 3).

Reviewer 2 Report

The revisions that the authors made has improved the article. Further minor revisions should be done before the article gets accepted for publication.

1.) Typo in Line 38: "Ngugen"

2.)  Please rewrite lines 40-42. It's as if the fish has the enzymatic and antioxidant activity. Did you mean that the enzymatic and antioxidant activities IN the fish were affected? 

3.) "," is missing in Line 69.

4.) Please rewrite Lines 90-92, "being clarify"

5.) Lines 107-109: Please rewrite. It reads as if you exposed different organisms to copper.

6.) Line 176: Spell out "2". 

7.)  Lines 244-246: Please rewrite. You used a conditional statement and coupled it with "were observed". I think it's better to say: will be observed or is expected to be observed. If this is not what you meant, just change the sentence.

8.) The discussion for energy metabolism and transport is very thorough. Please try to concise this part.

9.) Lines 337-384: Are scallops mammals? They are mollusks. And the discussion in this part is mostly associated with mammals. Any evidence that the same thing happens in the studied organism? Please use references that are, at the very least, related to scallops/other marine invertebrates. Mammalian physiology and invertebrates are different.

10.)  Lines 400-402: Please rephrase. you may write "upon exposure to 82 ug/L of copper for 48 hours"

11.) Lines 442-449: PGs in mammals/humans. The current study is on scallops.

12.) and may be, for example, linked...Please check your "," throughout the article.

13.) Line 608. The or then?

Author Response

Thank you very much for your comments and your thorough reading of the manuscript. We have changed all the sentences containing typographical and tense errors or inappropriate vocabulary. A thorough reading has been carried out to improve the punctuation. We have also modified the discussion as you suggested. We have reduced the section on energy and transport metabolism. And we have modified all the parts not referring to invertebrate metabolism either by replacing them with more appropriate examples or by deleting them when this was not possible.

Thank you very much for your time and the kindness of your answers.

  1. Typo in Line 38: "Ngugen"

Done

  1. Please rewrite lines 40-42. It's as if the fish has the enzymatic and antioxidant activity. Did you mean that the enzymatic and antioxidant activities IN the fish were affected? 

Yes this is what we mean. I was added in the sentence.

  1. "," is missing in Line 69.

Done

  1. Please rewrite Lines 90-92, "being clarify"

Done

  1. Lines 107-109: Please rewrite. It reads as if you exposed different organisms to copper.

Done

  1. Line 176: Spell out "2". 

Done

  1. Lines 244-246: Please rewrite. You used a conditional statement and coupled it with "were observed". I think it's better to say: will be observed or is expected to be observed. If this is not what you meant, just change the sentence.

Done

  1. The discussion for energy metabolism and transport is very thorough. Please try to concise this part.

The paragraph about “energy metabolism and oxygen transport” has been shortened by deleting some details considered less important for the discussion.

  1. Lines 337-384: Are scallops mammals? They are mollusks. And the discussion in this part is mostly associated with mammals. Any evidence that the same thing happens in the studied organism? Please use references that are, at the very least, related to scallops/other marine invertebrates. Mammalian physiology and invertebrates are different.

Whole sections of this paragraph have been rewritten to refocus the discussion on bivalves.

  1. Lines 400-402: Please rephrase. you may write "upon exposure to 82 ug/L of copper for 48 hours"

Done

  1. Lines 442-449: PGs in mammals/humans. The current study is on scallops.

The reference to mammals has been deleted as the information given is valid for bivlaves.

  1. and may be, for example, linked...Please check your "," throughout the article.

Done

  1. Line 608. The or then?

“Then” was replaced by “the”

Reviewer 3 Report

This manuscript represents much improvement and clarity to the presentation of this study on copper response to scallops. A few minor changes still remain and are detailed below.  

  • Line 558 watch link for parameters format  (didn't merge right in pdf) 
  • Table 1 is now legible but still really small and some of the lines didn't format properly in the pdf. Hopefully that can be improved during copyediting. 
  • Spectra for score 1 identifications should be presented 
  • Figure 2 presentation should be improved. There are overlapping plots & plots are low resolution. The entire figure didn't fit when the pdf merged. It is also visually a bit tricky to connect name with plot. Perhaps a line connecting name to plot might help or starting name closer to plots rather than scallop.  
  • For the figure 2 name groupings- the logic supplied in the response document should also be added to the text to clarify which plots represent single molecules and which are classes and how those groupings were chosen. 
  • I am still a little confused on sample numbers 84 vs 90? (13 control * 3= 39 + 15*3= 45) What happened to the other 2 control groups (6 scallops) to equal 90. These groupings as described in response document should be included in text. 
  • Their discussion in the response document of why 48 hours selected is great and should be included in manuscript (referencing their previous work). 
  • For the amount of copper chosen, yes it is in the intro, but would be helpful to remind readers all the way down in the methods where that number came from. (that's asking the reader to remember a lot and to implicitly figure out the significance of the study) 

Author Response

I would like to thank you for the accuracy of your comments and for the time you took to review our manuscript. All your comments have been taken into consideration to improve the content of our article. Further details on the questions you asked are included below. We hope that these corrections will meet your expectation.

  1. Line 558 watch link for parameters format (didn't merge right in pdf).

Indeed, the link is too long to be read in pdf format. Unfortunately, we cannot shorten it.

  1. Table 1 is now legible but still really small and some of the lines didn't format properly in the pdf. Hopefully that can be improved during copyediting. 

After several attempts on different computers, we do not have the same problem as you concerning the readability of table and figures 1 and 2 once in pdf format. They appear clear and complete. We will send a folder with all the figures in the most readable format possible as a supplement to the article. This should be enough to make them readable in the article. Figure 1 can be rotated (landscape layout) to enlarge it a little more, but this makes it uncomfortable to read.

  1. Spectra for score 1 identifications should be presented. 

The spectra corresponding to the analyzed standards were added in the materials and methods, as well as a sentence explaining the way we analyzed the standards (lines 579-582).

  1. Figure 2 presentation should be improved. There are overlapping plots & plots are low resolution. The entire figure didn't fit when the pdf merged. It is also visually a bit tricky to connect name with plot. Perhaps a line connecting name to plot might help or starting name closer to plots rather than scallop.  

Figure 2 has been modified as suggested to make it more readable.

  1. For the figure 2 name groupings- the logic supplied in the response document should also be added to the text to clarify which plots represent single molecules and which are classes and how those groupings were chosen. 

As rightly suggested, the explanations for the groupings of names have been given in the text (lines 595-598).

  1. I am still a little confused on sample numbers 84 vs 90? (13 control * 3= 39 + 15*3= 45) What happened to the other 2 control groups (6 scallops) to equal 90. These groupings as described in response document should be included in text. 

You are right, there is an error in the values but not in the calculation. The experiment starts with 13 exposed samples and 13 controls. However, during the analysis, samples were removed as they were considered outliers by the PCA (lines 136-140). 

Clarification of the number of individuals per condition, the number of samples and a sentence justifying the removal of these outlier samples has been added to the text (lines 490 - 501).

  1. Their discussion in the response document of why 48 hours selected is great and should be included in manuscript (referencing their previous work). 

As suggested, we added the explanation of the exposure time in the materials and methods section of the article.

  1. For the amount of copper chosen, yes it is in the intro, but would be helpful to remind readers all the way down in the methods where that number came from. (that's asking the reader to remember a lot and to implicitly figure out the significance of the study) 

Indeed, in order to improve the text comprehension, the context of the study has been added throughout the text.

This manuscript is a resubmission of an earlier submission. The following is a list of the peer review reports and author responses from that submission.

Round 1

Reviewer 1 Report

This is a review of the manuscript entitled “Untargeted metabolomics reveals a complex impact on different metabolic pathways in scallop Mimachlamys varia (Linnaeus, 1758) after short-term exposure to copper at environmental dose” as submitted to Metabolites. Below are some questions for enhancing the impact of this manuscript.

  1. Author need to explain the rationale of this study based on previous studies that investigated copper toxicity in terms of metabolomics, too much description for the port and its pollution in the introduction part. In the discussion part, previous studies have been well explained copper-induced toxicity underlying molecular mechanism. What is the novelty of this study, only different species?
  2. Author selected Mimachlamys varia as toxicological model and mentioned ‘it is very present in the Atlantic port areas. Then, how much of the biomolecular mechanism such as genes, transcripts, proteins, and metabolites of this animal model is revealed? Author should provide advantages of Mimachlamys varia in toxicological and, in particular, metabolomics study.
  3. In the section 2.1, metabolites had larger CV value in QC samples (>0.3). How to determine this value?
  4. In Figure 1, the scores plots explained below 30%, and clusters showed much uncertainty because experimental outliers existed outside of 95% confidence limit of Hoteling’s T2 statistic. Author should consider whether these models were well organized or not.
  5. Where VIP value was derived from? Whether the appropriate samples were used in PLS-DA or not? Author should change the format of Table 1 (too small) and show more figures to explain the metabolomics analysis.
  6. In the section 2.2, Shymanski scale was used for scoring of identified metabolites. How did metabolites were scored as 1 without any reference standard?
  7. Overall, more careful consideration is required to use of “identified” metabolites.
  8. In the discussion, there are too many metabolic alterations to clear understand comprehensive molecular mechanism of copper. What is the main toxic mechanism of copper and how they were connected?

Author Response

To all reviewers:
We would like to thank all the reviewers for their time and numerous comments. All these corrections have helped us to improve our article both in content and in form. We hope that we have been able to incorporate their comments and that this article will meet your expectations.

Reviewer 1:
1. Author need to explain the rationale of this study based on previous studies that investigated copper toxicity in terms of metabolomics, too much description for the port and its pollution in the introduction part. In the discussion part, previous studies have been well explained copper-induced toxicity underlying molecular mechanism. What is
the novelty of this study, only different species?

The introduction has been completely revised to highlight the ecotoxicological and metabolomic contributions of this study. The advantages of Mimachlamys varia for metabolomics study have been highlighted more explicitly in the text (see next point). Although previous studies have already been carried out on copper-induced toxicity in bivalves, using untargeted metabolomics approaches, we are far from the identification of all unexpected metabolic changes caused by copper in these organisms. Indeed, given the variety of chemical classes and physical properties that characterize
metabolites, a wide range of analytical techniques are required to achieve full coverage of the metabolome. The changes observed are dependent on both the extraction and analytical methods used. The present study therefore allows the discovery of new significant metabolic shifts in copper-exposed molluscs. We have added in the introduction part and in conclusion sentences to emphasize the importance of the choice of extraction and analytical methods for obtaining new results in metabolomics.

2. Author selected Mimachlamys varia as toxicological model and mentioned ‘it is very present in the Atlantic port areas. Then, how much of the biomolecular mechanism such as genes, transcripts, proteins, and metabolites of this animal model is revealed? Author should provide advantages of Mimachlamys varia in toxicological and, in particular,
metabolomics study.

Mimachlamys varia is a commonly used bioindicator on the Atlantic coast, especially for studies on trace elements such as copper. However, there are no studies showing the effects of copper on the molecular mechanisms of scallops. It is to fill some of these gaps that we have chosen to work on this species. The current state of knowledge on this species has been added in the introduction. The interest of this species as a study model was also underlined in the introduction. (Lines 69-85)

3. In the section 2.1, metabolites had larger CV value in QC samples (>0.3). How to determine this value?

Typically, the coefficient of variation (CV) is calculated across pooled QC samples for each feature and those with a CV above a predetermined cutoff (e.g., 20–30%) are removed. Thresholds of 20-30% are classic in this type of study, the maximal CV tolerated is usually 0.3 (see below 3 references): 1°) Giacomoni F, Corguille GL, Monsoor M, Landi M, Pericard P, et al. Workflow4metabolomics: A collaborative research infrastructure for computational metabolomics. Bioinformatics. 2015;31(9):1493–5. doi: 10.1093/bioinformatics/btu813, 2°) Thevenot EA., Roux A., Xu Y., Ezan E., and Junot C. (2015). Analysis of the human adult urinary metabolome variations with age, body mass index and gender by
implementing a comprehensive workflow for univariate and OPLS statistical analyses. Journal of Proteome Research, 14:3322-3335 (http://dx.doi.org/10.1021/acs.jproteome.5b00354), 3°) Reinke Stacey N., Gallart-Ayala Héctor, Gómez Cristina, Checa Antonio, Fauland Alexander, Naz Shama, Kamleh Muhammad Anas, Djukanović Ratko, Hinks Timothy
S.C., Wheelock Craig E. Metabolomics analysis identifies different metabotypes of asthma severity. European Respiratory Journal. 2017;49(3):1601740. doi: 10.1183/13993003.01740-2016.

4. In Figure 1, the scores plots explained below 30%, and clusters showed much uncertainty because experimental outliers existed outside of 95% confidence limit of Hoteling’s T2 statistic. Author should consider whether these models were well organized or not.

Indeed, the first 2 components of the score plots explained a low level of variability. However, considering the very high number of variables involved in the sample variability, it is logical that the first 2 axes of the score plot represent a low variability. Nevertheless, the sample clustering appears clearly for both ionization modes.

5. Where VIP value was derived from? Whether the appropriate samples were used in PLSDA or not? Author should change the format of Table 1 (too small) and show more figures to explain the metabolomics analysis.

The VIP is the Variable Importance on Projection. It is a value used as a numerical indicator of importance of each ion, each X-variable in the model of PLSDA. It is calculated for each variable based on weighed sum of squares of PLS-DA, weighted according to the Y variation accounted for by each component. As the mean of all VIP is 1, values above 1 characterize ions with a significant contribution to the model. The highest the VIPs are, the more important are the variables in the model. Significant changes have been made to table 1 to make it more readable. In order to give more details on the analysis performed, the link to the workflow used is added to the publication. Concerning your second question about appropriate sample use: as specified in the manuscript, we performed PLS-DA after excluding outliers detected previously thanks to PCA Hotelling T2.

6. In the section 2.2, Shymanski scale was used for scoring of identified metabolites. How did metabolites were scored as 1 without any reference standard?

We have rectified this and reclassified the ions with previous score 1 and without analytical standards.

7. Overall, more careful consideration is required to use of “identified” metabolites.

Indeed, only metabolites that have been compared to a standard can be considered as identified. We have therefore moderated our assertions for the others.

8. In the discussion, there are too many metabolic alterations to clear understand comprehensive molecular mechanism of copper. What is the main toxic mechanism of copper and how they were connected?

A full paragraph was added at the beginning of the discussion in response to this comment: “One of the most significant challenges, when performing environmental untargeted metabolomics study, is identifying which metabolic pathways are altered, in relation with the metabolite variations observed in stressor-exposed organisms. Many metabolites are involved in several pathways and are the product or substrate of many different enzymes or processes, in particular for central metabolic pathways where any one metabolite play a role in a myriad of pathways. The translation from changes in metabolites relative abundance to physiological interpretation is therefore a key issue. Nevertheless, in the following discussion, we propose different hypotheses on the physiological effects connected with the identified metabolites that present a significant modulation after copper exposure. At this stage, it is not possible to establish connections between the potentially involved metabolic pathways, nor to prioritize the effects. It is clear that these hypotheses, based on previous works, need further experiments to be supported, but they show the complexity of copper toxicity mechanisms and provide a solid starting point for further biological interpretation.”

Reviewer 2 Report

The study is well-written, coherent, and comprehensible. However, here are some points to consider to improve the study:

1.) Please indicate the concentration of copper that was found to be deleterious to marine life in your introduction.

2.) I like how creative Figure 2 is but please change this to a visualization that is more appealing. What's the unit of the y-axis?

3.) Are these significantly expressed metabolites quantified and identities verified (with standards)? Are the physiological effects stated in the study, as a response to the upregulation/downregulation of the metabolites tested empirically? Or are the claims merely based on previous works?

4.) Implications of the study are missing (the template in the discussion was this: if this metabolite is upregulated/downregulated, then this metabolism is affected-->and then?)

Untargeted metabolomics is indeed an effective tool in exploring the effects of different compounds on various organisms, however, we should note that this is only a part of the whole picture. Changes in the relative abundances of metabolites do not really prove anything about the physiology and metabolism of the organism, hence, further experiments (gene expression, absolute quantification of metabolites) are warranted to support these claims. 

The article reads more like a review rather than an original article.

Author Response

To all reviewers:
We would like to thank all the reviewers for their time and numerous comments. All these corrections have helped us to improve our article both in content and in form. We hope that we have been able to incorporate their comments and that this article will meet your expectations.

Reviewer 2:
The study is well-written, coherent, and comprehensible. However, here are some points to consider to improve the study:
1. Please indicate the concentration of copper that was found to be deleterious to marine life in your introduction.

The concentrations for which copper is considered deleterious to marine life have been added in the introduction (lines 36-42). Throughout the manuscript, quantitative information has been added where possible.

2. I like how creative Figure 2 is but please change this to a visualization that is more appealing. What's the unit of the y-axis?

In order to make figure 2 more appealing and to improve comprehension a legend has been added.

3. Are these significantly expressed metabolites quantified and identities verified (with standards)? Are the physiological effects stated in the study, as a response to the upregulation/downregulation of the metabolites tested empirically? Or are the claims merely based on previous works?

Only leucine and aspartic acid could be compared to standards. The others were annotated according to the agreement between the spectral data and the literature. Where possible, fragments of the ions were used to improve the identification of the parent ions. The work here is semi-quantitative, i.e. the relative intensity of the ions by comparing an averaged relative intensity with 15 samples per condition. An absolute quantification would not, in our opinion, bring anything more since the homeostasis of the scallop is  not known. In our opinion, the "copper modulation" already reflects the impact of the metal and the associated stress effect and illustrates which metabolic pathways are most affected. Our article does not aim to explore all the metabolic pathways of the scallop and to measure each metabolite, but rather to determine whether there is a stress linked to a specific pollution and to identify the metabolic pathways affected. This exploratory work is based on data available in the literature only.

4. Implications of the study are missing (the template in the discussion was this: if this metabolite is upregulated/downregulated, then this metabolism is affected-->and then?)

Untargeted metabolomics is indeed an effective tool in exploring the effects of different compounds on various organisms, however, we should note that this is only a part of the whole picture. Changes in the relative abundances of metabolites do not really prove anything about the physiology and metabolism of the organism, hence, further experiments (gene expression, absolute quantification of metabolites) are warranted to support these claims.

The conclusions have been completely rewritten to answer to these comments:

“This study constitutes a broad survey on the metabolites impacted by copper exposure in scallops, at environmental concentration (82 μg/L). We used an optimized triple extraction method and as analytical technique, ultra-high performance liquid chromatography coupled to high resolution mass spectrometry (UHPLC-HRMS), to detect as many metabolites as possible, without a priori. However, given the variety of chemical classes and physical properties that characterize metabolites, it is clear that not all the different chemical classes of metabolites could be covered in a single analytical method. Then significant varying metabolites between non-exposed and exposed organisms have been isolated using multivariate statistics and among them, 30 have been identified. The result is that we have a partial view of copper-induced metabolic shifts. Under these conditions, we can only put forward hypotheses on the metabolic pathways affected, which remain to be confirmed. Furthermore, at this stage, it is not possible to establish connections between the different metabolic pathways, nor to prioritize the effects and to elucidate the complete molecular mechanism of copper toxicity, the complexity of which is already highlighted by our results. Besides these limitations of an untargeted metabolomics approach, this study demonstrates some of the strengths of using this method: (i) the detection of 'distress signals' at the molecular level, faster than in organism/population/ecosystem level biomarkers (ii) the identification of metabolic changes that would be missed by other 'omics' techniques, as protein and mRNA levels in the cell do not necessarily translate into metabolite changes (iii) the detection of unexpected varying metabolites, showing that short exposure to copper at environmental condition leads to significant changes in the metabolism of scallops, some of which are signs of cellular damage. Further studies are needed to formally identify the key metabolic pathways involved in copper toxicity. Tools are available which attempt to confirm assertion made for altered pathways through (i) subsequent focus on key pathways of interest with a targeted metabolomics analysis, (ii) an enrichment analysis, (iii) the building of net-works to help interpret the data, (iv) isotopic labelling a substrate and following this label through various metabolic pathways.”

The article reads more like a review rather than an original article.

We hope that the major corrections made to this article will highlight the contribution of our work and give you satisfaction.

Reviewer 3 Report

The authors have conducted an interesting but simple study comparing the metabolomes of port scallops exposed to copper compared to no treatment. However potentially interesting, the illegible table and incomplete data presentation, results description and methods section mean that the manuscript could not be properly evaluated at this time. With revisions that provide complete results, methods and data, the scientific merit of this publication can be evaluated. 

Specific comments, but revisions should not be limited to the few things mentioned here. 

Abstract and introduction:

  • There are phrasing issues throughout these sections. I can decipher what the authors mean, but they have chosen atypical words, used words that lack scientific or quantitative rigor or not included the necessary comparison or detail following an adjective (examples but not all instances are listed below).  
  • overuse of the word 'very'. Very is not quantitative and not useful unless compared to something. 
  • line 46-  phrases like 'become deleterious' are not correct. Deleterious to what? genes? fish? How are they being deleted? Do the authors mean toxic or deadly? 
  • line 51- phrases like 'most present' are not quantitative or accurate. Higher concentration compared to what other trace elements? Most agencies have threshold values for allowable limits. How do the values detected compare to regulatory levels? 
  • line 57- watch for typos. A space is missing in between the city and country  
  • line 59- watch presentation of units throughout.  Here a period does not belong after ug 
  • A missing piece of information is what is known about regular metabolism of these scallops? Only response to copper is filter feeders is discussed. Has the typical metabolome been profiled? 
  • Pertubations are typically called dysregulated features in the non-targeted metabolomics community 

Methods (section 4)

4.1 

  • Why was 48 hours of exposure selected? How do the authors know this is enough time to create a phenotypic response? 
  • How was the amount of copper chosen? 
  • Why were scallop gills harvested and not the full tissue? Are the gills representative of total metabolism? 
  • If 15 samples were analyzed for exposed and not exposed = 30. What happened to the other 15 from the original 45? How were the 30 analyzed chosen? 

4.2-4.4 

  • For each of these sections, much more methods details are needed. Even if you are following a previous publication, you need to briefly describe what you have done include all instrument run solvents, modes, column, times, mass spec parameters and software parameters and analysis work flows including processing, stats, identification (which is really annotation because not MS2 data for all but 1) and scoring. 
  • Line 450: LCMS not LCMS/MS was used to identify? What conditions was this run on? Where was the standard obtained from? Where are the spectra verifying the formal identification. 
  • Line 452- how is abundance determined? I did not see a calibration curve. Do they just mean spectral intensity? 

Results (Section 2)

  • All figures and tables suffer from clarity issues. Table 1 had to be zoomed in to over 300% and I still couldn't read it properly. Figure 1 the dotted line does not appear as dots. Figure 2- meaning of y-axes and how calculated is not useful and changes from plot to plot. Also n=? stats method? error bars or significance method?  Some of the metabolites are names as a single molecule and some as a group of molecules like trisaccharides- is that 1 trisaccharide feature that couldn't be formally identified or all such sugars together?
  • In the results, the experiments should be briefly summarized before describing the results, especially since the methods are presented last. You cannot assume that readers know what all the different scores are and what experiment you conducted to obtain the different pieces of data. Results need to be described within their experimental and computational analysis context. 
  • 2.1 should start with a summary of the work including the sampling and growth of the scallops. No data is given on the phenotype or growth of the scallops. How did they know the scallops were acclimated after 1 month? What parameters had stabilized?  Were all the scallops behaving similarly? How was age or size of the scallops controlled? How long had the scallops been growing in the port before harvesting? How was the species of scallop identified? 
  • Issues with presenting values with sub or superscripts ie: R2 not R2 
  • further evaluation of results and discussion was not possible because of issues with legibility and missing information 

Author Response

To all reviewers:
We would like to thank all the reviewers for their time and numerous comments. All these corrections have helped us to improve our article both in content and in form. We hope that we have been able to incorporate their comments and that this article will meet your expectations.

Reviewer 3:
The authors have conducted an interesting but simple study comparing the metabolomes of port scallops exposed to copper compared to no treatment. However potentially interesting, the illegible table and incomplete data presentation, results description and methods section mean that the manuscript could not be properly evaluated at this time. With revisions that provide complete results, methods and data, the scientific merit of this publication can be evaluated. Specific comments, but revisions should not be limited to the few things mentioned here.

Major corrections have been made to the whole document; the introduction has been thoroughly modified to highlight the interest of this study. The results have been clarified and important details have been added to the material and method. A complete paragraph was added at the beginning of the discussion and the conclusion was completely rewritten. We hope that these major corrections will satisfy you and bring more clarity to our article.

Abstract and introduction :
There are phrasing issues throughout these sections. I can decipher what the authors mean, but they have chosen atypical words, used words that lack scientific or quantitative rigor or not included the necessary comparison or detail following an adjective (examples but not all instances are listed below).

A thorough and careful review of the whole document has been carried out to correct this.

1. overuse of the word 'very'. Very is not quantitative and not useful unless compared to something.

Indeed, the word "very" was used excessively. To remedy this, we chose to modify all the sentences that contained this adverb and we have added quantitative data where possible.

2. line 46- phrases like 'become deleterious' are not correct. Deleterious to what? genes? fish? How are they being deleted? Do the authors mean toxic or deadly?

To clarify understanding an unambiguous vocabulary has been used.

3. line 51- phrases like 'most present' are not quantitative or accurate. Higher concentration compared to what other trace elements? Most agencies have threshold values for allowable limits. How do the values detected compare to regulatory levels?

Clarifications have been made and the word "most" has been removed from the whole text. In addition, quantified data and effective comparisons have been provided where possible. The regulatory data are based on daily discharge rates and not on concentrations in the environment, which makes them difficult to compare. For this reason, we have chosen not to display these regulatory values.

4. line 57- watch for typos. A space is missing in between the city and country

A thorough reading of the document for typos was carried out to avoid such errors.

5. line 59- watch presentation of units throughout. Here a period does not belong after ug

To simplify reading all units written in the format "x.y-1" have been replaced by the format "x/y".

6. A missing piece of information is what is known about regular metabolism of these scallops? Only response to copper is filter feeders is discussed. Has the typical metabolome been profiled?

Our objective is not to study the typical scallop metabolome, but to identify significant metabolite shifts in copper-exposed scallops compared to non-exposed ones, using an untargeted metabolomic approach. Indeed, this method allows the measurement of the widest range of metabolites present in an extracted sample without prior knowledge of the metabolome of the studied organism.

The state of the art of current knowledge on this species was added in the introduction to clarify this point (lines 64-85).

7. Pertubations are typically called dysregulated features in the non-targeted metabolomics community

As rightly suggested by the reviewer the word "perturbations" is replaced by the word "dysregulated features".

Methods (section 4)

4.1
1. Why was 48 hours of exposure selected? How do the authors know this is enough time to create a phenotypic response?

We wanted to observe the effects of copper after the longest possible exposure, but before lethal effects occur. In a pre-test study, we observed high mortality in scallops after 72 hours of exposure to copper at the same concentration (82μg/L), while the controls were still alive. We therefore chose to study the effects of copper (82μg/L) on the metabolism of scallops one day before the lethal effects occurred. This choice is reinforced by a study showing that copper can have lethal effects in some bivalve species after 48 h of exposure to concentrations below our own:

Species Exposure Time  Chemical Concentration Effect
Anodonta anatine
Pseudanodonta
complanata Unio
tumidos
48 hours  Copper LC50 = 18.9
(10.0–31.1) μg/L
LC50 = 29.3
(25.4–34.0) μg/L
LC50 = 19.0
(16.2–22.0) μg/L
Lethality

(From Mesquita, A. F., Marques, S. M., Marques, J. C., Gonçalves, F. J., & Gonçalves, A. M. (2019). Copper
sulphate impact on the antioxidant defence system of the marine bivalves Cerastoderma edule and
Scrobicularia plana. Scientific reports, 9(1), 1-11. and coming from Kováts, N., Abdel-Hameid, N. A., Kovács,
K. & Paulovits, G. Sensitivity of three Unionid glochidia to elevated levels of copper, zinc and lead. Knowl.
Manag. Aquat. Ecosyst.399, 04, https://doi.org/10.1051/kmae/2010028 (2010).)

2. How was the amount of copper chosen?

As stated in the introduction, the chosen copper concentration is 82 μg/l because it corresponds to the maximal level found in 2018 in the largest marina in Europe. This allows us to study the effect of this element on the metabolism of the organisms living in this ecosystem.

3. Why were scallop gills harvested and not the full tissue? Are the gills representative of total metabolism?

A paragraph has been added to justify the choice of gills as the collected organs with two supplementary references:
“Gills in bivalves such as scallops were enlarged during evolution and are much more extensive than needed for respiration. They form the main interface between the organism and the surrounding water and have become therefore a key organ for food absorption. Given the high water filtration rates of bivalves, gills constitute also a
significant pathway of incorporation of pollutants and among them metals via seawater (Ravera 2001). Making a logical extension to this, it has been shown that gills would have high defense ability against contaminants, which are characterized by induction of pollutant biotransformation and antioxidant-related enzymes and metallothioneins
(Trevisan et al., 2016). Consequently gills play a central role in scenarios of acute exposure to metals, by integrating both absorption and metabolism.”

4. If 15 samples were analyzed for exposed and not exposed = 30. What happened to the other 15 from the original 45? How were the 30 analyzed chosen ?

Indeed, a calculation error slipped into the text, it is not 45 scallops in total but 45 scallops per condition that is 90 scallops in total. This corresponds to 15 replicates per condition (control and exposed) with 3 scallops per replicate.

4.2-4.4

5. For each of these sections, much more methods details are needed. Even if you are following a previous publication, you need to briefly describe what you have done include all instrument run solvents, modes, column, times, mass spec parameters and software parameters and analysis work flows including processing, stats, identification (which is really annotation because not MS2 data for all but 1) and scoring.

As rightly mentioned by the reviewer, the technical elements of the analysis have been added to the protocols to improve understanding.

The following explanation has been added, concerning the MS and MS/MS experiments: “The MS and the MS/MS experiments were performed using the MSE function (Waters) in centroid mode. A MSE approach consists in MS and MS/MS data acquisitions in a single same run. This is achieved by rapidly alternating between two functions i.e., the
first, acquired at low energy provides exact mass precursor ion spectra; the second, at elevated energy provides high energy exact mass of the fragment ions. Fragment ion spectra are assigned to their associated precursor ion peaks so that all the information necessary to identify each compound of interest is collated and available. Software algorithms that profile each chromatographic peak and determine their retention times accomplish this. Precursors and fragment spectra are then aligned according to retention times and linked together.”

6. Line 450: LCMS not LCMS/MS was used to identify? What conditions was this run on? Where was the standard obtained from? Where are the spectra verifying the formal identification.

We have detailed in the previous section the whole spectrometry method with the MSE approach which consists of a simultaneous acquisition of MS and MS/MS data. The supplier of standards was added. The same analysis is carried out on the standards.

7. Line 452- how is abundance determined? I did not see a calibration curve. Do they just mean spectral intensity?

It is indeed the spectral intensity and not an abundance. This misuse of language is corrected throughout the text.
Results (Section 2)

8. All figures and tables suffer from clarity issues. Table 1 had to be zoomed in to over 300% and I still couldn't read it properly. Figure 1 the dotted line does not appear as dots. Figure 2- meaning of y-axes and how calculated is not useful and changes from plot to plot. Also n=? stats method? error bars or significance method? Some of the metabolites are names as a single molecule and some as a group of molecules like trisaccharides- is that 1 trisaccharide feature that couldn't be formally identified or all such sugars together?

All figures have been modified to improve understanding. The table 1 has been simplified and enlarged. The lines in the figure 1 have been redesigned. A legend and the number of statistical individuals has been added to the figure 2. However, the box plots are current graphical representations of statistical data. They do not require a statistical method and follow norms (i.e., they represent the median, quartiles, minimum and maximum of a sample). As we use "classical" boxplots (without notch transformation or other) we do not consider it necessary to explain the structure.

In Figure 2, the scale of y-axes is changed between graphs for better readability. In fact, there is a factor 10 between some graphs, if they were all aligned on the same scale, some would not even be visible. The aim here is not to compare these boxplots with each other, but to have for each ion an idea of the effect of copper on their relative
spectral intensity.

Here, it is difficult to say whether several structures are co-eluted, because trisaccharides of different structure have the same formula and therefore the same m/z. Finally, the same fragment ions will be observed. So, we have grouped these types of metabolites under the same generic name.

9. In the results, the experiments should be briefly summarized before describing the results, especially since the methods are presented last. You cannot assume that readers know what all the different scores are and what experiment you conducted to obtain the different pieces of data. Results need to be described within their experimental and computational analysis context.

A paragraph was added at the end of the introduction part to summarize the experimental and analytical context of the study.

10. 2.1 should start with a summary of the work including the sampling and growth of the scallops. No data is given on the phenotype or growth of the scallops. How did they know the scallops were acclimated after 1 month? What parameters had stabilized? Were all the scallops behaving similarly? How was age or size of the scallops controlled?How long had the scallops been growing in the port before harvesting? How was the species of scallop identified?

A supplementary part was added in the materials and methods (see 4.1): "The experimental design is based on an experiment conducted by Ory and collaborators [16]. For the present study, 90 scallops (size between 4-6 cm) were collected at La Pointe du Grouin in Loix en Ré along the French Atlantic coast (Ré Island, France) [...]".

All the scallops are measured during sampling (4-6 cm as mentioned in the material and method). Scallop growth is not sufficiently fast to be significantly altered in 4 weeks and the number of samples used (45 per condition) is large enough to consider that inter-individual variability is negligible.

The scallops did not grow in the harbour as this could compromise our analyses. Indeed, we want to study the effect of copper only, and scallops grown in the harbour would have been stressed by other trace elements. Our scallops come from a reference site, identified for its low trace element content. To clarify this point, a paragraph has been added to the introduction (lines 110-113).

“A genetic approach was applied in a previous study to validate the identity of the species collected at this reference site. This study also confirmed that the scallop M. varia along the French Atlantic and English Channel coasts is not genetically subdivided. Consequently, the scallops found at the refence site are comparable to those found in the port of Les Minimes.” This information is added in the material and methods (lines 481-491).

11. Issues with presenting values with sub or superscripts ie: R2 not R2

This typo has been corrected.

Further evaluation of results and discussion was not possible because of issues with legibility and missing information
We hope that all the corrections made will meet your expectations, fill in the missing information and make the document more readable.